# Identifying the Hidden Population: Former Intravenous Drug Users Who Are No Longer in Contact with Services. “Ask a Friend”

**DOI:** 10.3390/diagnostics11020170

**Published:** 2021-01-25

**Authors:** Sarah R. Donaldson, Andrew Radley, John F. Dillon

**Affiliations:** 1School of Medicine, University of Dundee, Dundee DD1 9SY, UK; andrew.radley@nhs.scot (A.R.); j.f.dillon@dundee.ac.uk (J.F.D.); 2Directorate of Public Health, NHS Tayside, Dundee DD3 8EA, UK; 3Department of Gastroenterology, Ninewells Hospital & Medical School, Dundee DD1 9SY, UK

**Keywords:** hepatitis C, models of care, PWID, social networks, respondent-driven sampling, dry blood spot

## Abstract

People who, after a period of drug use, have changed their lifestyle and left substance use behind them are a hidden population within our communities. Lack of contact with drug services may mean that they are not tested for hepatitis C (HCV) infection through service-led initiatives and, therefore, may be exposed to the chronic morbidity and risk of death inherent with a legacy of HCV infection. This study utilized respondent-driven sampling (RDS) in a novel fashion to find those at historical risk of HCV. The social networks of people with a history of drug use were mapped, and individuals not currently in contact with services were invited to come forward for testing by members of their social network. The study used a reference group to inform study methodology and communication methods to reach out to this hidden population. One hundred and nine individuals received dry blood spot tests for HCV, 17.4% were antibody positive. Fifty one individuals met the inclusion criteria for this study. One hundred and twenty three invite-to-test coupons were issued; however, only one wave of recruitment consisting of one participant resulted from this method. Using RDS in historical social networks was not effective in this study and did not reach this hidden population and increase testing for HCV. This study is registered with clinicaltrials.gov (Ref NCT03697135).

## 1. Introduction

National Health Service (NHS) Tayside, a health board area in Scotland, has targeted significant resources over the past 7 years to test, treat and care for people with hepatitis C (HCV). Tayside is recognised as having outstanding testing and treatment rates, and the region recently announced that it has reached Scottish 2024 elimination targets four years earlier than anticipated [1,2,3]. However, the population that has moved beyond substance use has thus far been hidden and out of reach of the novel treatment pathways developed and is recognised as an important but complex group that is difficult to reach [3]. A number of novel studies have successfully targeted people who have a history of substance use to improve the health of each individual and to utilise treatment as prevention to improve the health of the community [3,4,5,6,7,8].

Health behaviours and lifestyles are shaped by a person’s social network [9,10,11,12]. People socialise with others who hold similar views, values and lifestyles, where certain behaviours are the norm [13]. Social networks can provide support and guide positive lifestyle choices, improving the health of the individual and providing a downstream effect that influences the health behaviours of other group members [10,11]. However, social networks may also influence negative lifestyle choices and create a barrier to recovery [10]. Successful recovery from substance use is associated with changing your social network [12,14,15].

The lack of current contact of this hidden population with drug services may mean that they are not tested for hepatitis C (HCV) infection through service-led initiatives and therefore they may be exposed to the chronic morbidity and risk of death inherent in HCV infection. The introduction of efficient testing methods [16] as well as highly effective oral treatments for HCV means that identifying and treating these individuals has become a clinical priority; they should be offered treatment before the effects of their hepatitis infection becomes irreversible. There remains a question of how to effectively reach this population who view any risk for HCV as being left far behind in their past.

Social networks have been identified as both route of transmission of blood borne viruses (BBV) and a route to reach individuals for harm reduction, testing and treatment [17]. Previous studies have demonstrated the effectiveness and efficiency of using members of a social network to recruit fellow members of their shared community [18,19]. Using a respondent-driven sampling (RDS) technique, the social networks of people with a history of drug use are mapped and individuals not currently in contact with services are encouraged to come forward for testing and potential treatment [20]. This method has been demonstrated to be an effective methodology to recruit people who inject drugs in the UK for dry blood spot testing (DBST) for HCV [21].

The key difference in this study is that we aimed to use this chain referral method to reach beyond current social networks, tapping into historical networks to reach those who have moved on from substance use.

## 2. Materials and Methods

### 2.1. Reference Group

In order to investigate the likely feasibility of this approach, a reference group was established with participants who had lived experience of substance use or had worked in drug services. The group consisted of eight people with lived experience and four people who worked within drug services. The group met within a local care centre providing drug and recovery services. The group provided many supportive examples of how people who previously injected drugs were known to live within local communities and provided the research team reassurance that current contacts within services could identify the target population.

The reference group was used to provide participant input into the study design and implementation. The reference group supported the design of study documents to ensure they were relevant and valid. The group advised that risk from tattoos should be included as inclusion criteria. Advice was sought from the reference group regarding providing monetary incentives to support recruitment for this study. The group provided the opinion that the potential participants for the study were likely to be in a better financial position due to their recovery journey and so the altruistic behaviour would outweigh any financial incentives of a feasible scale. Therefore, this study did not use monetary incentives to recruit.

### 2.2. Recruitment Method

The study utilised RDS not as a tool to estimate prevalence of HCV in this population but in a highly novel fashion to find those at historical risk of HCV. An initial group of participants, recruited by the researchers (known as seeds), provided the first wave of this method. These seeds were then asked to recruit from their social network (a further group known as alters) to provide recruitment for the second wave. This process continued until saturation [18,22,23]. We estimated that we would require 25 initial seeds of former or current injecting drug users within the Dundee community to reach a target audience of 500 through two waves of recruitment with up to five invite-to-tests per seed.

### 2.3. Study Procedure

DBST was offered to individuals identified as having risk factors for HCV by members of the research team from community-based hubs and drop in services in Tayside. DBST followed standard operating procedures and guidance prepared by the Sexual Health and Blood Borne Virus Managed Care Network [24]. DBST were carried out according to the method described by Judd et al. with local validation [25].

The individual was provided with information about the HCV test and offered referral for treatment, if required, as NHS standard of care. Patient information sheets (PIS) were provided to the individual at the time of the DBST, and they were advised of the opportunity to join the study. If an individual declined to participate they followed the usual pathway of care.

The individual was advised of the date when the test result would be ready, and an agreement made regarding where and how they would like to receive the results. At the point of receiving results, the individual was asked if they would like to participate in the study and provide consent, according to standard practice. At this point, participants became seeds to begin waves of recruitment of alters.

Seed participants were asked to identify members of their previous injecting network by providing general information about these individuals to create a pseudo-identity. A mapping tool was used to assist in identifying these individuals (Figure 1). A standard dataset was collected: given name, approximate age (if known). The pseudo-identities created by the seed participants were insufficient to identify an actual person but enabled the seed participant to focus on whom they might contact.

Advice was provided to the seed participants about sensitively approaching individuals based on feedback from the reference group. They were advised to only approach individuals that they felt confident and safe approaching, only approach individuals in safe places and to not start discussions if the individual was unwilling to engage.

Seed participants were issued with study coupons sufficient to cover the number of potential alters identified by them. Each study coupon contained a unique reference number that linked it to the seed who gave the alter the invitation. The seed participants were also issued with study information cards to explain the study to those they attempted to recruit.

The alters identified by seeds represented Wave 1 of the respondent-driven sample. Seed participants were contacted weekly by phone or when visiting the community hubs and drop-in’s for up to four weeks, by a designated member of the research team to ask about progress in contacting social network contacts.

The people handed coupons by the Wave 1 participants constituted Wave 2 participants and so on, following the same method as described above.

### 2.4. Inclusion Criteria

To be eligible for the study, participants were aged 18 years or over with an identified risk for HCV infection from previous or current history of injecting drug use or tattoos. Participants had to be willing to identity and contact previous injecting partners after training, according to study procedures, and to provide informed consent.

## 3. Results

A total of 109 individuals received DBST as part of the standard pathway of care in Tayside at community hubs, drop in’s and recovery cafes between August 2018 and December 2019. Researchers visited eight different sites during this 16-month period; all sites provide support for people who use substances or are in recovery from substance use. The study sites visited were the Cairn Centre (run by Hillcrest Futures), the signpost centre drop in (run by We Are With You), Cocaine Anonymous group (run by We Are With You), Making Dundee Home, Main Street Cafe, Stobswell Advice Cafe, Dundee Parish Nursing at the Steeple and the Connect Café. All eight sites were located in Dundee City in Tayside. The researchers attended each site between four and eleven times with a total of 54 visits, depending on DBST uptake and volume of attendance. Fifty of the individuals that accepted DBST did not return for their results despite numerous attempts by the researchers to find them and so were not eligible to be recruited to deliver the study protocol. Fifty One individuals met the inclusion criteria and provided consent to participate in the study to map social (former drug using) networks and invite contacts for testing. The study profile is outlined in Figure 2.

Of the 109 individuals offered DBST, 82.6% were recorded as antibody-negative (*n* = 90) and 17.4% as antibody-positive (*n* = 19). Of the 19 antibody-positives, 47.4% were PCR-negative (*n* = 9), 26.3% were PCR-positive and required further care (*n* = 5, including two who declined treatment), 21.1% required further testing to establish PCR test results (*n* = 4) and 5.3% (*n* = 1) undergoing treatment. Eight individuals returned for DBST results but were ineligible for the study as they did not disclose a risk for HCV from previous or current drug use or from tattoos.

Participant characteristics are described in Table 1 below.

Compared to Table 1, data for Scotland in 2015/16 showed that 71% of people who experienced problematic substance use were male and the 35–64 year age range had the highest number of individuals at 64%, 25–34 years 26% and 15–24 years 10% [26]. The Scottish Drugs Misuse Database (SMR25) shows that less than 1% of clients report ethnicity as something other than white [27].

### 3.1. Participant Risk Factors for HCV

The reference group was very clear that risk from tattoos should be an inclusion criterion as many were obtained in risky environments. The reference group also advised that tattoos may be an acceptable cover for other risks that the participant may feel uncomfortable disclosing due to the associated stigma, such as injecting drug use. 66.7% (*n* = 34) of participants identified they had a tattoo, and 9.8% (*n* = 5) of participants reported tattoo as the only risk factor. 67.6% (*n* = 23) of participants obtained the tattoo in environments where sterile equipment may not have been used.

### 3.2. Networks and Invites

Table 2 shows drug using networks and the number of invite-to-test coupons issued and returned. 47.1% (*n* = 24) of participants mapped their social network of people they had used drugs with. 45.8% (*n* = 11) of these had a network of between three and five people with whom they had previously used drugs with. Three participants reported that they had used drugs with a high number of people; this increased the mean to 13.7. Only one participant reported that they currently used drugs with someone else (between three and five people). Fourteen participants reported that they could identify people at risk but declined to use the mapping tool to describe the pseudo-identities but were willing to use the coupons as an invite-to-test.

One hundred and twenty three invite-to-test coupons were issued with 74.5% of participants reporting they were happy to give them to people within their social network. The average number of coupons issued was 3.2; guided by how many people the seed thought that they could reach. The number of coupons returned to obtain a DBST was one.

## 4. Discussion

This study found a low incidence of antibody-positive DBST results (17.4%) for this group with current or historical risk factors for HCV, in the recovery cafes, drop-in’s and recovery communities that were visited. The prevalence rate of positive HCV antibodies for people who inject drugs in Tayside was estimated to be 56% in 2018 [28]. This might reflect that our study was recruiting from areas where there was low prevalence of drug use; however, 84.3% of seeds reported current or historical drug use and the figure we reported is higher than the estimated prevalence for antibody-positive results from HCV testing from all sources in Tayside (5.1%) [29]. This suggests that the study sites were appropriate venues for recruitment.

NHS Tayside is recognised for its outstanding testing efforts [3]. The low prevalence of antibody positives in this group suggests that significant progress made in Tayside to identify those at risk of HCV meant that there were few cases left to find in this group. The group accepting DBST at recovery cafes, hubs and drop in’s may have been more willing to engage with testing due to a recovery journey that encompasses progress towards overall health, wellbeing and future plans [30]. Conversely, those at high risk may not have been attending these groups, or unfortunately the consequences of risk behaviours may have been realised.

### 4.1. Return for Test Results and Seed Recruitment

The study found a high proportion of individuals that accepted the offer of a DBST did not return for the test results (45.9%). HCV is a stigmatised disease due to its close links with injecting drug use [31,32,33]. For those that have made steps to move beyond substance use, it may be an unwelcome reminder of a past life. Fear of the test result, consequences of disclosure to family and friends and the associated stigma might have influenced return rates.

However, despite this the study recruited a larger number of initial seed participants (*n* = 51) than the estimated 25 to reach a target audience of 500. The estimate made an assumption that recruiting using RDS methodology would act in the same way in historical networks as current user networks and that two waves with between 3 and 5 invite-to-test coupons per seed would reach our target [18,19,21]. Only 74.5% (*n* = 38) of the initial seeds took invite-to-test coupons, which might indicate that the coupon design was not acceptable to some participants’ despite review of study materials by the reference group.

It is also notable that a large number of individuals who wished to be tested for HCV did not attend for results. This is despite plans put in place by the research team to meet them in their communities and perseverance through repeated attempts.

### 4.2. Reference Group Advice on Monetary Incentives

The reference group was made up of people with lived experience and those working in drug services and as such had in-depth knowledge and experience in this field. Many RDS studies use financial incentives as a strategy of maintaining seed engagement with recruitment to strengthen other motivating factors of peer pressure and receiving care [18,34]. The study did not use monetary rewards as it was advised by the reference group that this was not necessary, as the targets of recruitment were likely to be in a better financial position and so the altruistic behavior would outweigh any feasible financial incentives proposed. The view of our reference group was that receiving a test is the primary reason for engagement. Using monetary or non-monetary incentives has not been associated with success or failure of a study [35,36]. Indeed some studies warn of ethical considerations in utilising a monetary incentive that may encourage coercion or duress and an underground diversion of coupons [36,37].

### 4.3. Social Networks and Waves

Despite recruiting 51 initial seeds, 38 of whom took invite-to-test coupons, the study only recruited one alter. Data collected showed that initial seeds could identify few active injecting drug users within their social networks but could identify a mean of 13.7 individuals within historical networks that they believed they still had links to. The study settings and the social network mapping suggests that the seeds recruited were in recovery, as few could map active users but had not yet moved completely beyond substance use and could still identify former social contacts.

### 4.4. Success of Peer-to-Peer Invite-to-Test

The study demonstrates that this attempt to utilise peer-to-peer invitation to test was not a successful method to cross social network boundaries and reach those who have left substance use behind.

### 4.5. Limitations of the Study

There are a number of limitations to the study. A high proportion of male participants recruited may have influenced the success of onward recruitment, with gender potentially affecting willingness to recruit others. Other studies have found little difference recruiting across gender lines when recruiting those with a history of drug use [23,38]. The research team consisted of one male and one female and so addressed potential researcher bias in selecting gender to approach for recruitment.

The study recruited participants who reported high use of stimulants, which reflects a growing trend in Scotland [28]. This may be due to our attendance at a Cocaine Anonymous group where nine participants were recruited from. Individuals who use stimulants may not identify with the typical drug-using identity and therefore may be harder to reach using this method.

It is unclear if including a monetary reward would have encouraged this target population to come forward for a test despite the advice from our reference group that it was not necessary.

## 5. Conclusions

Our attempt to use RDS to identify historical social networks was unsuccessful. Individuals engaging with testing for HCV may be sensitive to the associated stigma and, potentially, to unwelcome reminders of past history when they have moved beyond substance use. The seed participants found it hard to reach beyond their current social networks in community hubs and drop-in’s. It is unclear if monetary incentives would be effective for this population, and the scale of the incentive required to overcome the barriers identified is also unclear. Further research is required to identify novel approaches to reach this target population to test and cure HCV before the consequences of chronic disease are realised. Consideration should be given to exploring motivating factors for getting a HCV test for this hidden population. Establishing a reference group with participation from members who do not attend services may provide additional insights. However, the difficulty recruiting a reference group from this population should not be underestimated.

## Figures and Tables

**Figure 1 diagnostics-11-00170-f001:**
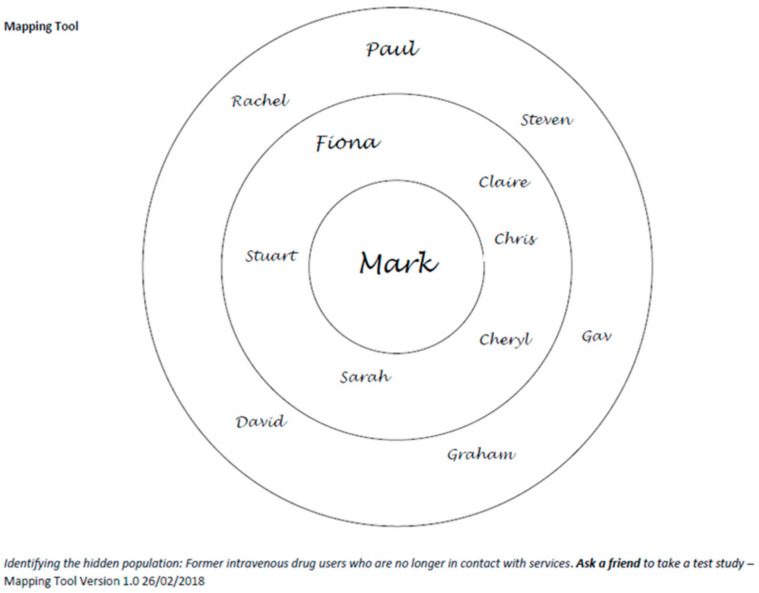
Social network mapping tool example. The participants name is placed in the centre, the inner circle represents individuals with whom the participant has a close relationship with and/or is in contact with on a regular basis. The outer circle represents individuals the participant regards as being within their social network but that they are in contact less frequently and/or are considered to be acquaintances rather than friends.

**Figure 2 diagnostics-11-00170-f002:**
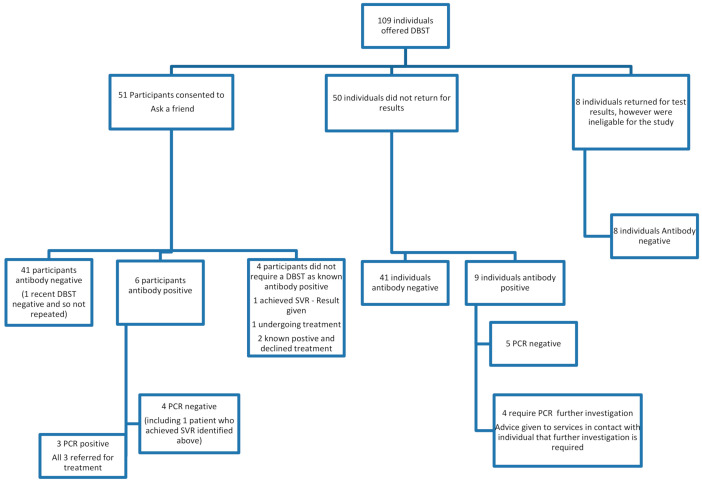
Study profile.

**Table 1 diagnostics-11-00170-t001:** Participant characteristics.

(*n* = 51)	Percentage
Male	80.40%
Female	19.60%
Age	
>45 years	45.10%
35–44 years	35.30%
25–34 years	13.70%
<25 years	5.90%
Ethnicity	
White British or white other	94.10%
Ethnicity not disclosed	5.90%
Participants who reported to have taken drugs in the past or present	84.3% (*n* = 43)
Drug use reported (*n* = 43)	
Heroin	39.5% (*n* = 17)
Other opiates	25.6% (*n* = 11)
Methadone *	30.2% (*n* = 13)
Buprenorphine *	30.2% (*n* = 13)
Cocaine	65.1% (*n* = 28)
Benzodiazepine (not prescribed)	46.5% (*n* = 20)
Ecstasy	55.8% (*n* = 24)
Cannabis	86.0% (*n* = 37)
Steroids	13.9% (*n* = 6)
Psychedelics	23.3% (*n* = 10)
Amphetamines	9.3% (*n* = 4)
Novel psychoactive substances (NPS)	2.3% (*n* = 1)

* may be prescribed as opiate substitution therapy (OST).

**Table 2 diagnostics-11-00170-t002:** Network and invite-to-test results.

	Number of Participants	Percentage of Participants
Number of seeds who use the mapping tool (Figure 1) to map their historical network		(*n* = 51)
	24	47.1%
Number of people in social network seeds historically used drugs with		(*n* = 24)
≤2	3	12.5%
2–5	11	45.8%
6–10	7	29.2%
11–20	0	0.0%
21–50	1	4.2%
51–100	1	4.2%
>100	1	4.2%
Mean number of people in social network seeds historically used drugs with *	13.7	+/− 10.9 (95% CI)
Number of people seeds currently use drugs with		(*n* = 51)
≤2	1	2.0%
≥2	0	0.0%
Invite-to-test coupons		(*n* = 51)
Number accepting	38	74.5%
Number declined to take	13	25.5%
Number of invite-to-test coupons issued to seed		(*n* = 38)
100.0%	3	7.9%
200.0%	6	15.8%
300.0%	18	47.4%
400.0%	2	5.3%
500.0%	8	21.1%
>5	1	2.6%
Total number of invite-to-test coupons issued	123	
Mean number of invite-to-test coupons issued to seeds	3.2	+/− 0.4 (95% CI)
		(*n* = 123)
Number of invite-to-test coupons returned for a test	1	0.8%

* Where participants identified 3 or 4, the number was taken to be the lower estimate.

## Data Availability

The data presented in this study are available on request from the corresponding author. The data are not publicly available due to privacy.

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
