# Peer review of "Identifying the Hidden Population: Former Intravenous Drug Users Who Are No Longer in Contact with Services. “Ask a Friend”"

_diagnostics, 2021, doi:10.3390/diagnostics11020170_

Round 1
Reviewer 1 Report
Donaldson et al. have used a novel recruitment strategy to try and identify people who used to inject drugs, that are no longer engaged with services - an extremely important population to reach if HCV elimination is to be achieved.
The paper is well written, the results and interpretation are clear. There are some minor suggestions the authors may like to consider to improve readability of the manuscript. These are described below:
Abstract Line 18 - however only one (check spelling?) wave of recruitment consisting of one participant resulted from this method.
Line 79-82 - formatting query - paragraphs / returns?
Some additional subheaders under section 2 may be helpful? Eg. ethics, reference group, recruitment method, study procedure
Would it make sense for the feedback and advice from the Reference group to be described as a result of the paper? Eg. justification of decision not to include monetary incentives.
Figure one - Mapping tool. What do the different co-centric circles mean? Significance of location of in each one, size etc? This Figure would be improved by adding some more detail / explanatory text
Methods - more detail on the study design, eg. recruitment period, location of 8 sites would be helpful.
Line 160 - estimates of #s required, should this be in methods / analysis section?
Figure 2
- barcode description - note described in text (does this equal the identifying number?) Check for consistent terminology throughout.
- A more descriptive title could be considered. Eg. Study profile?
- Is it worthwhile describing why 8 individuals ineligible
- Clarify in diagram, that 4 participants that did not require DBS as known Ab positive? Add PCR negative for the individual who achieved SVR
Line 171-176 - comparison with expected prevalence, and lines 179- 180 re comparison with Scotland # should this be in discussion? Likewise Line 198 - This high figure is likely to be due to our attendance at a Cocaine Anonymous group where nine participants were recruited from….
Line 180 - is another world other than “problem” drug users possible here?
Line 188 - “cover for other risks” - does this mean a surrogate factor for other risks?
Line 192 onwards - would any of these be worthwhile including in the Table 1 patient characteristics?, eg. reported taking drugs in past or present?
Table 2 - Suggest adding more detail to the column headers to enable to enable the reader to understand independently of text? Suggest adding the demoninator in the table ot make this clear.to make it clear. ot make this clear. Look at formatting of bottom of table - is this out of line?
Do the authors have suggestions on other methods to improve recruitment from this hidden population? Are there any suggestions of reflections from the reference group, or others that could / be consulted to explore further.
Author Response
Point 1: Donaldson et al. have used a novel recruitment strategy to try and identify people who used to inject drugs, that are no longer engaged with services - an extremely important population to reach if HCV elimination is to be achieved.
The paper is well written, the results and interpretation are clear. There are some minor suggestions the authors may like to consider to improve readability of the manuscript. These are described below:
Response 1: We thank the reviewer for their kind comments and greatly appreciate them taking the time to do this.
Point 2: Abstract Line 18 - however only one (check spelling?) wave of recruitment consisting of one participant resulted from this method.
Response 2: Line 18 spelling error amended from “on” to “one”
Point 3: Line 79-82 - formatting query - paragraphs / returns?
Response 3: Line now 84-85 formatting corrected.
Point 4: Some additional subheaders under section 2 may be helpful? Eg. ethics, reference group, recruitment method, study procedure
Response 4: Additional sub headers added to the text.
Line 66 – Ethics
Line 72 - Reference group
Line 98 -Recruitment method
Line 109 - Study procedure
Line 154 - Inclusion criteria
Point 5: Would it make sense for the feedback and advice from the Reference group to be described as a result of the paper? Eg. justification of decision not to include monetary incentives.
Response 5: The section discussing monetary incentives in RDS has been moved from the methods section and is included in a subheading as part of the discussion on lines 283-296.
Amendments have been made to the remaining text describing the reference group in the methods section on lines 84-93.
Point 6: Figure one - Mapping tool. What do the different co-centric circles mean? Significance of location of in each one, size etc? This Figure would be improved by adding some more detail / explanatory text
Response 6: Additional information has been added to figure one (lines 133-137) to describe the co-centric circles
Point 7: Methods - more detail on the study design, eg. recruitment period, location of 8 sites would be helpful.
Response 7: Details of the recruitment period and the location of the 8 sites have been added to the manuscript on lines 162-170.
Point 8: Line 160 - estimates of #s required, should this be in methods / analysis section?
Response 8: Estimate of numbers required moved to the methods section (lines 104-107) and the wording slightly amended.
Point 9: Figure 2
barcode description - note described in text (does this equal the identifying number?) Check for consistent terminology throughout.
A more descriptive title could be considered. Eg. Study profile?
Is it worthwhile describing why 8 individuals ineligible
Clarify in diagram, that 4 participants that did not require DBS as known Ab positive? Add PCR negative for the individual who achieved SVR
Response 9: Identification numbers (referred to as barcodes) were allocated to study paperwork to manage the issuing of invite-to-test coupons. The box describing 111 barcodes issued and 2 invalid in a separate box have been removed from figure 2 as on reflection they refer to the management of study material and not study profile.
Diagram amended to clarify that 4 participants did not require DBS as they were known antibody positive. PCR negative added to figure to include individual who achieved SVR in the box above.
*Please note that the changes have been highlighted in red in this figure and not as track changes as they appeared in black ink*
A description of why the 8 individuals were ineligible has been added to lines 185-187.
Point 10: Line 171-176 - comparison with expected prevalence, and lines 179- 180 re comparison with Scotland # should this be in discussion? Likewise Line 198 - This high figure is likely to be due to our attendance at a Cocaine Anonymous group where nine participants were recruited from….
Response 10: Lines 171-176 and 179-180 comparing prevalence and comparison with Scotland removed. These figures are already described in the discussion section lines 245-252. An addition was made to this section of the discussion on lines 245-246. Line 198 regarding attendance at a Cocaine Anonymous group has been moved to the discussion section (lines 320-321).
Point 11: Line 180 - is another world other than “problem” drug users possible here?
Response 11: Problem drug use is the terminology used in the official statistics publication for Scotland. The authors agree that using “problem drug users” is not a desirable phrase to use and have changed the terminology to people first language on lines 197-198.
Point 12: Line 188 - “cover for other risks” - does this mean a surrogate factor for other risks?
Response 12: Additional information has been added to describe that participants may feel more comfortable disclosing a risk from a tattoo rather than a risk from injecting drug use due to stigma on lines 205-207.
Point 13: Line 192 onwards - would any of these be worthwhile including in the Table 1 patient characteristics?, eg. reported taking drugs in past or present?
Response 13: Participant characteristics relating to drug use have been included in table 1 and the descriptive text relating to drugs used on lines 212-224 removed.
Point 14: Table 2 - Suggest adding more detail to the column headers to enable to enable the reader to understand independently of text? Suggest adding the denominator in the table to make this clear. Look at formatting of bottom of table - is this out of line?
Response 14: Table 2 has been amended to include further details to allow independent understanding from the text. The denominator has been added at each section. The formatting issue has been addressed
Point 15: Do the authors have suggestions on other methods to improve recruitment from this hidden population? Are there any suggestions of reflections from the reference group, or others that could / be consulted to explore further.
Response 15: Additional reflections have been added to the conclusion section lines 337-346 to suggest that future research could explore monetary incentives in this population and exploration of other motivating factors. A reflection to include members of this population in future reference groups could provide additional insights.
Reviewer 2 Report
The authors presented the usefulness of DRS to find the hidden HCV-infected population in order to screen more patients who can enter clinical care to facilitate HCV elimination. Generally, the manuscript was well-written and did provide important information about the policy making for these special clinical setting.
- Please provide the reasons of the 8 subjects who were ineligible for the study
- Table 2 information should be refined. The rows in the table should be well matched.
- Typos throughout the manuscript.
Author Response
Point 1: The authors presented the usefulness of DRS to find the hidden HCV-infected population in order to screen more patients who can enter clinical care to facilitate HCV elimination. Generally, the manuscript was well-written and did provide important information about the policy making for these special clinical setting
Response 1: We thank the reviewer for their kind comments and greatly appreciate them taking the time to do this.
Point 2: Please provide the reasons of the 8 subjects who were ineligible for the study
Response 2: A description of why the 8 individuals were ineligible has been added to lines 187-189.
Point 3: Table 2 information should be refined. The rows in the table should be well matched.
Response 3: Table 2 has been amended to include further details to allow understanding independent from the text. The denominator has been added at each section. The formatting issue has been addressed.
Point 4:Typos throughout the manuscript.
Response 4: Typos in the manuscript have been addressed.